# Landsat-8/9 Atmospheric Correction Reliability Using Scene Statistics

**David Groeneveld [1,*], Tim Ruggles [1] and Bo-Cai Gao [2]**

[1] Advanced Remote Sensing Inc., Hartford, SD 57033, USA
[2] Remote Sensing Division, Naval Research Laboratory, Washington, DC 20375, USA; bo-cai.gao@nrl.navy.mil
[*] Correspondence: david@advancedremotesensing.com

**Abstract:** Landsat data correction using the Land Surface Reflectance Code (LaSRC) has been proposed as the basis for the atmospheric correction of smallsats. While atmospheric correction can enhance smallsat data, the Landsat/LaSRC pathway delays output and may constrain accuracy and utility. The alternative, the Closed-form Method for Atmospheric Correction (CMAC), developed for smallsat application, provides surface reflectance derived solely from scene statistics. In a prior paper, CMAC closely agreed with LaSRC software for correction of the four VNIR bands of Landsat-8/9 images for conditions of low to moderate atmospheric effect over quasi-invariant warehouse-industrial targets. Those results were accepted as surrogate surface reflectance to support analysis of CMAC and LaSRC reliability for surface reflectance retrieval in two contrasting environments: shortgrass prairie and barren desert. Reliability was defined and tested through a null hypothesis: the same top-of-atmosphere reflectance under the same atmospheric condition will provide the same estimate of surface reflectance. Evaluated against the prior surrogate surface reflectance, the results found decreasing error with increasing wavelength for both methods. From 58 comparisons across the four bands, the LaSRC average absolute error ranged from 0.59% (NIR) to 50.30% (blue). CMAC provided reliable results: error was well constrained from 0.01% (NIR) to 0.98% (blue).

**Keywords:** surface reflectance; retrieval; LaSRC; CMAC; scene statistics; near real time; spectral diversity

## 1. Introduction

Through smallsats, electro-optical Earth observation (EO) is rapidly expanding, enabled by advances in electronics, imaging sensors, data transmission, and the miniaturization of components. The resulting smallsat constellations provide rapid repeat imagery that is needed to better understand and manage the unprecedented planetary-scale threats from climate change. Perhaps the greatest challenge for all EO data applications is that the data are obtained through the atmosphere, which variably corrupts the data. The solution is to correct the data to surface reflectance, a process that seeks to remove the atmospheric effect entirely, resulting in clear images and restored digital signals. However, atmospheric correction is problematic for the many hundreds of EO smallsats without onboard equipment to calibrate sensor output, permitting direct conversion to surface reflectance.

A proposed pathway for smallsat surface reflectance correction applies the Land Surface Reflectance Code (LaSRC) and cross calibration to the data of two research grade satellite platforms, Landsat-8/9 and Sentinel-2 A/B [1,2], atmospherically corrected by LaSRC. Such cross-calibration can introduce uncertainty due to mismatched overpass timing and spectral responses between sensor platforms. LaSRC currently requires ancillary data to assess the degree of atmospheric effect on the day of the smallsat's image acquisition, both for calibration and application. This potentially adds another layer of uncertainty in operational surface reflectance retrieval due to any temporally mismatched image collection for the ancillary data. Ancillary data also delay image processing and output and may have

coarser granularity that reduces spatial sensitivity. CMAC was formulated to avoid these sources of uncertainty.

The Closed-form Method for Atmospheric Correction (CMAC) was developed to deliver surface reflectance with no delay upon image download. Its development was prompted by a seminal observation of atmospherically driven reflectance changes. The novel CMAC pathway was tested against Sen2Cor for Sentinel-2 correction [1] and LaSRC for Landsat-8/9 correction [2]. Corrected Landsat data by LaSRC, L2A, are available only through Earth Explorer and represent the general current state of the art in surface reflectance retrieval. CMAC proved accurate and precise for higher levels of atmospheric effect, estimated from each image's spectral data alone. This paper investigates an additional inquiry into CMAC application: whether the reliability of surface reflectance output from one area of interest (AOI) applies to all other environments, especially those with very different spectral characteristics. As a yardstick for this comparison, CMAC reliability is compared to the state-of-the-art LaSRC correction for the four visible and near infrared bands (VNIR) of Landsat-8/9.

The paucity of surface reflectance data is a challenge for evaluating atmospheric correction. A few such datasets exist, but virtually never in time and space to support sustained, focused testing to compare methods, thus necessitating a workaround for investigating the reliability of atmospheric corrections. The workaround for this investigation relies upon a truism relative to atmospheric correction: correction accuracy is greatest for scenes taken through a relatively "clean" atmosphere. This is partially because clean images require much less adjustment to achieve surface reflectance but also because the engineering tolerances to accurately retrieve surface reflectance become tighter and the solution accuracy is more critical as atmospheric effects increase. In a prior investigation [2], CMAC and LaSRC closely agreed on images acquired under relatively clean atmospheric conditions. These data were accepted as surrogate surface reflectance to provide a reference to construct the datasets to support this analysis. These surrogate surface reflectance estimates were consulted to find TOAR reflectance values present in both data sets that could be used to test the null hypothesis described below.

This paper (1) investigates the reliability of CMAC to provide surface reflectance output for environments widely different from where CMAC was developed and calibrated, and (2) evaluates the functionality of CMAC in relationship to the widely accepted state of the art, LaSRC. A null hypothesis was formulated to express atmospheric correction reliability: *Equivalent top-of-atmosphere reflectance from images of different environments collected under the same level of atmospheric effect, when atmospherically corrected, will yield equivalent surface reflectance (i.e., no difference).* This hypothesis can be stated more simply as "the same input affected by the same conditions will yield the same output".

A point of continual reference in past papers that we use again here is the application of image appearance that includes clarity and color balance of atmospherically corrected images [1,2]. Though a subjective interpretation, images that appear clear are logically closer to surface reflectance than those containing visible haze. Figure 1 is an example of an extremely hazy image portrayed as top-of-atmosphere reflectance (TOAR) and CMAC- and LaSRC-corrected views and illustrates the importance of scene appearance for judging atmospheric correction. While the lack of surface reflectance groundtruth in the appropriate time and at an appropriate scale can be problematic for cross-checking the accuracy of retrieved surface reflectance, image appearance is useful as a qualitative guide for correction accuracy.

CMAC represents several innovations. These include the use of scene statistics, alone, for retrieval of surface reflectance rather than dependence upon ancillary data from another satellite. Scene statistics provide assessment of the atmospheric effect as a lump sum in the form of a grayscale map, with brightness conveying a greater degree of correction. The resulting surface reflectance retrieval accounts for scatter and absorption using a conceptual model that inverts and adjusts the empirical line method [3]. Another innovation is the representation of the atmospherically induced deviation of TOAR from surface

reflectance in Cartesian space as a line. The slope and offset of the line are applied as the two parameters that reverse TOAR to deliver surface reflectance differentially for each pixel across the image.

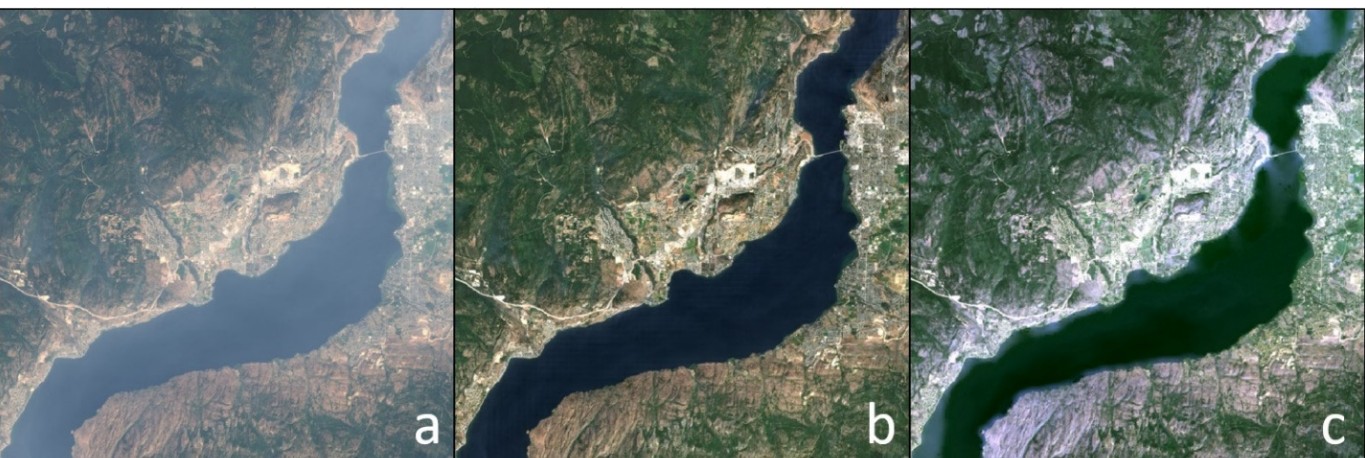

**Figure 1.** The quality of atmospheric correction can be judged by image clarity and color balance. Landsat-8 image of Kelowna, British Columbia, 8 August 2021, portrayed in true color: (**a**) uncorrected TOAR, (**b**) CMAC corrected, and (**c**) LaSRC corrected.

This paper does not specifically address solar angle or pointing direction, which inevitably lead to adjustment of the signal correction: these use cases await further investigation. However, given that the atmospheric effect is evaluated as a lump-sum "see it—correct it" approach, such effects may be partially controlled without additional consideration. Solar angle and path length through variable levels of atmospheric aerosol affect target irradiance, and this correction is planned to be approached through future empirical measurements.

## 2. Materials and Methods

CMAC is a recently developed method intended for atmospheric correction of smallsat images. Sentinel-2 data were used as the research and development testbed for CMAC. The process flow applies image reflectance in two steps that map and then reverse the atmospheric effect spatially.

The first step estimates the atmospheric effect based on the remarkably stable blue band reflectance properties of vegetation. Using a reference crop, alfalfa, and image extraction and sampling, the top-of-atmosphere reflectance (TOAR) was modeled based on the sampled VNIR spectral band responses. In application, the model is applied through grid sampling of VNIR spectral bands that predict the TOAR blue band response. Because the atmospheric model works with scene statistics, it supports surface retrieval in near real time without other inputs. When displayed as an image, the output of this initial step produces a grayscale whose brightness is applied to adjust the degree of correction.

In a second step, the CMAC processing reverses the atmospheric effect using a conceptual model that captures our observation of reflectance behavior under increasing aerosol loading: TOAR reflectance for dark targets increases due to backscatter and for bright targets decreases due to attenuation. This response is a bandwise linear continuum from dark to bright reflectance encoding the deviation from surface reflectance as a line for all pixels under a set atmospheric condition. The slopes and offsets of linear response are calibrated for each band of a satellite that then enables efficient reversal of the atmospheric effect to deliver the original surface reflectance. These steps and how they were developed are described in Appendix A, and the interested reader can also consult previous journal papers [1,2] for additional information.

This investigation applies an earlier comparison of CMAC to LaSRC (version LPGS_15.5.0) for 31 relatively clear Landsat-8 and -9 images of five AOIs of warehouse/industrial districts in Southern California (SoCal) known to have consistent surface reflectance [2]. In that analysis, the average cumulative distribution functions (CDFs) for these two disparate methods agreed to such a close extent that they plotted virtually atop one another (Figure 2). Those paired datasets are employed here as surrogate estimates of surface reflectance. As can be followed through an annotated spreadsheet in Appendix B, averaging and interpolating the values for CDF extractions from the 31-image cohort were used to construct datasets for CMAC and LaSRC comparison. This pairing found the same atmospheric conditions and the same TOAR input from completely different environments than SoCal, where these methods were in close agreement. The high dark-to-bright dynamic range of reflectance in the SoCal dataset is an important distinction, because the AOIs selected for comparison of method reliability have extremely low dark-to-bright dynamic spectral range.

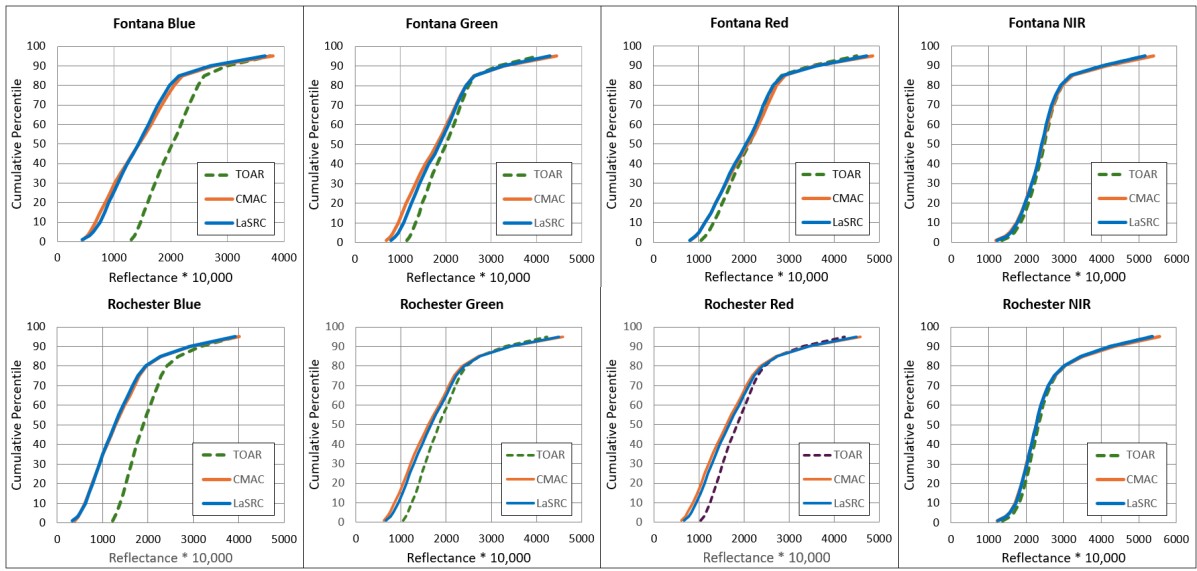

**Figure 2.** Bandwise reflectance CDFs for the four VNIR bands averaged for all 31 images of Landsat-8 and -9 of TOAR, CMAC, and LaSRC treatments extracted from AOIs in two Southern California municipalities. The resulting curves show nearly complete agreement between CMAC and LaSRC. High dynamic spectral range, dark to bright, for each band is demonstrated by the wide range of extracted reflectance values.

Landsat-8 and -9 data were not considered separately given the close agreement of these paired satellites [4]. LaSRC applies a radiative transfer (RadTran)-based workflow that is documented in readily attainable remote sensing literature that readers are urged to consult [5,6]. RadTran calculations account for the various reflectance, absorbance, transmittance, etc., components to estimate the amount of light and radiance measured by the sensor.

Two locations were selected for analysis of method reliability, both with extremely low dark-to-bright dynamic spectral range, very different from other areas where CMAC has been calibrated or applied. No specific selection criteria were considered for these two locations other than their low dark-to-bright spectral diversity being notably difficult for RadTran-based atmospheric correction. The first AOI investigated was located just west of Lake Newell, Alberta, Canada, and was chosen to represent shortgrass prairie, a vegetation cover occupying a band of semiarid climate that runs north to south for over 2000 km across the United States and Canada. After appreciating the results from Lake Newell, a second site adjacent to the El Pinacate volcanic uplands in Sonora, Mexico, was chosen for confirmation. El Pinacate represents a profound desert of exposed sand, with sparse, widespread shrubby trees constituting less than three percent cover within the AOI

investigated; such profound deserts are found in significant proportions of South America, Asia, Africa, and Australia. Shapefiles were mapped to enclose homogeneous cover for both AOIs (Figure 3). A regional view from the 4 July 2022 Landsat-8 El Pinacate region in Appendix C provides a wider view of the TOAR, CMAC-corrected, and LaSRC-corrected examples and further context for atmospheric correction over deserts with low dynamic spectral reflectance, contrasting with adjacent areas surrounding the Gulf of California shore exhibiting high dynamic spectral reflectance.

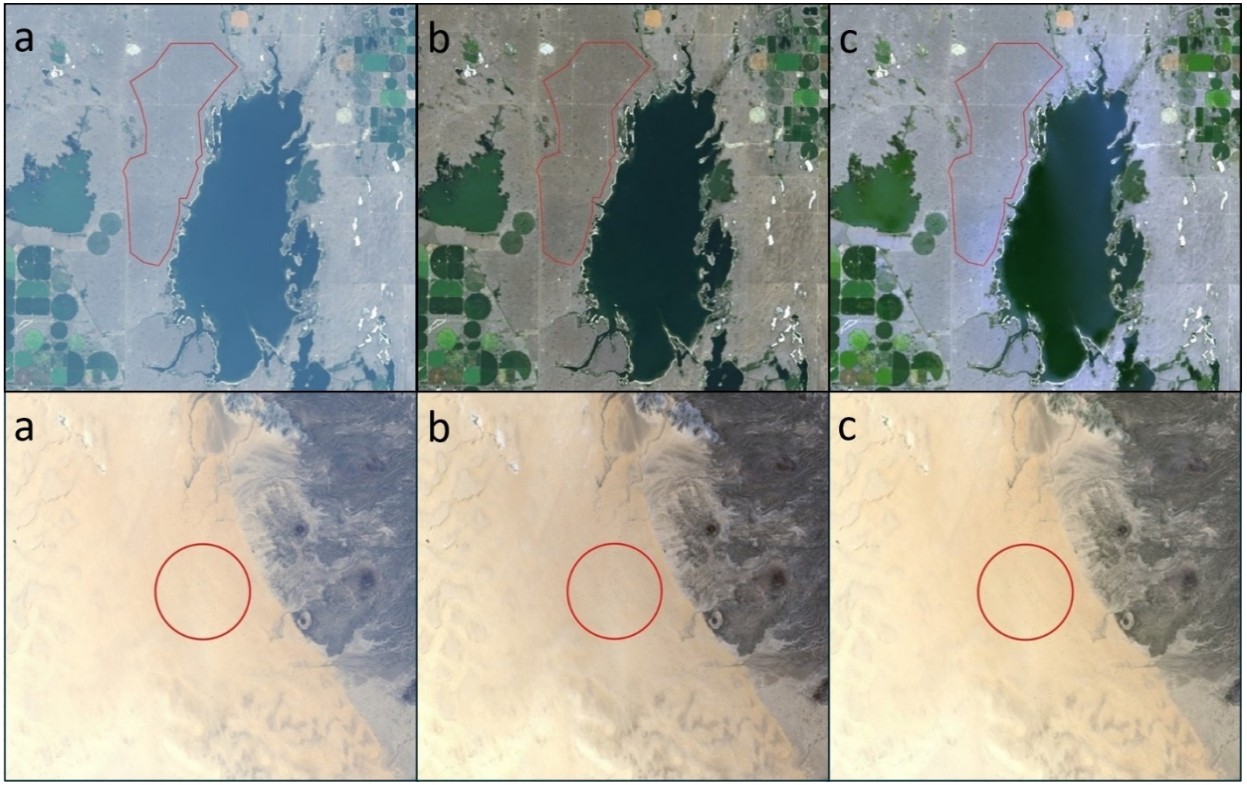

**Figure 3.** AOIs of Lake Newell (L8, 5 August 2023; **top**) and El Pinacate (L8, 4 July 2022; **bottom**) shown for three treatments: (**a**) TOAR; (**b**) CMAC; and (**c**) LaSRC. The area inside the Lake Newell AOI is 27.46 km$^2$ and 12.62 km$^2$ for the El Pinacate AOI.

Three Landsat-8 and -9 images were selected for each AOI from the Landsat archives and downloaded as both uncorrected and LaSRC-corrected (by version LPGS_15.5.0) images from Earth Explorer (Table 1). These six images were corrected by CMAC v1.1L and calibrated for Landsat-8/9 application; the same processing was also applied in the previous CMAC-to-LaSRC comparison [2]. The dataset to test the null hypothesis was constructed through back comparison to the SoCal datasets from AOIs with quasi-invariant reflectance, where CMAC and LaSRC showed close agreement. Any of the five SoCal AOIs would work for this application, because the 31 images were affected by a relatively moderate atmospheric effect, and because these two disparate methods resulted in virtually the same corrected data distributions per AOI. Two were selected, Fontana and Rochester, whose reflectance distributions are shown in Figure 2. Since the true surface reflectance is unknown, and there were slight differences between CMAC and LaSRC in the SoCal results, this analysis regarded the SoCal surface reflectance estimates as most appropriate for application as the atmospheric correction method: CMAC for CMAC and LaSRC for LaSRC. This ensured that the choice of datasets did not bias the results. For these comparisons, Atm-I was applied to measure the atmospheric effect rather than the aerosol optical thickness ancillary data applied in LaSRC from the Moderate Resolution Imaging Spectroradiometer (MODIS) [7]; however, the LaSRC datasets downloaded from Earth

Explorer were generated with the workflow that applies MODIS AOT for assessment of atmospheric effect [5,6].

**Table 1.** Data for the six Landsat-8 and -9 images that were selected for this analysis.

| AOI | Path/Row | Landsat | Image Date | Median Atm-I | Matched With | Match Atm-I Avg |
|------|----------|---------|------------|--------------|--------------|-----------------|
| Lk Newell | 040/025 | L9 | 7-29-2023 | 924 | Fontana | 924 |
| | 040/026 | L8 | 8-06-2023 | 1044 | Fontana | 1044 |
| | 040/027 | L9 | 8-14-2023 | 920 | Rochester | 920 |
| El Pinacate | 038/038 | L8 | 6-02-2022 | 965 | Fontana | 965 |
| | 038/039 | L8 | 7-04-2022 | 965 | Fontana | 965 |
| | 038/040 | L9 | 8-29-2022 | 983 | Fontana | 983 |

Pixel values for the blue, green, red, and NIR bands were extracted from within the AOI polygons shown in Figure 3 and exported to spreadsheets for analysis. Reflectance distributions from each image for three treatments (i.e., TOAR and corrected CMAC and LaSRC) were extracted and represented as CDFs in 21 percentiles from 1% to 3% and in 5-percentile steps between 5% and 95%. Data for the Rochester AOI were matched with the 8-14-2023 Lake Newell image to check for bias from selection of the SoCal AOI; none was found in comparison to data for the Fontana AOI, which was matched with the other five images (Table 1).

Construction of the dataset for testing the null hypothesis began by finding the Atm-I of multiple images of the SoCal datasets whose averages equaled the median Atm-I's from the images selected for Lake Newell and El Pinacate. The individual values of the 21 percentile steps for the distributions are arrayed in columns in the spreadsheets, one column per image, ranked by increasing Atm-I. This format facilitated averaging image values to support the comparisons by pairing the experimental image TOAR, CMAC, and LaSRC values with the corresponding averaged values for the SoCal images.

Appendix B provides portions of the combined 29 July 2023 Lake Newell and Fontana datasets that were reformatted and annotated to support explanation of the workflow to identify values for the three defining properties of the null hypothesis: (1) Atm-I conditions in the SoCal dataset were selected whose averages equaled the experimental datasets, thereby achieving the same atmospheric conditions; (2) interpolation to identify the exact TOAR and its percentile position in each of the six experimental images to match the SoCal TOAR input for atmospheric correction; and (3) identifying the corresponding atmospherically corrected output values from the SoCal dataset for testing the null hypothesis. Interrelating the TOAR data and the surface reflectance calculated from this data was accomplished within each dataset using their percentile positions. The 7-29-2023 Lake Newell spreadsheet contained in the Supplementary Materials can be compared to Appendix C to assist in following the calculation workflow.

The error for the atmospherically corrected data was estimated by treating the Fontana- and Rochester-corrected surface reflectance estimates as the standard to assess CMAC and LaSRC error: % error = $100 \times$ (value − standard)/standard. This comparison was judged to be valid because the atmospheric correction results for the spectrally diverse SoCal AOIs were accepted as surrogate surface reflectance. The "value" in this formula represents the Lake Newell and El Pinacate surface reflectance estimates.

This statistical distribution-based workflow was repeated for all four bands for each Lake Newell and El Pinacate image. In this manner, a series of common TOAR values, and the CMAC and LaSRC surface reflectances estimated from them, were interpolated from these two datasets.

## 3. Results

Spreadsheets, software, image lists CDFs for the four bands of the three treatments of the three images per AOI afford a comprehensive look at the responses per correction method (Figure 4). The TOAR CDFs for Lake Newell illustrate diverse reflectance due to the vegetated shortgrass prairie in comparison to El Pinacate, where the reflectance remained consistent for the ground surface virtually devoid of perennial vegetation.

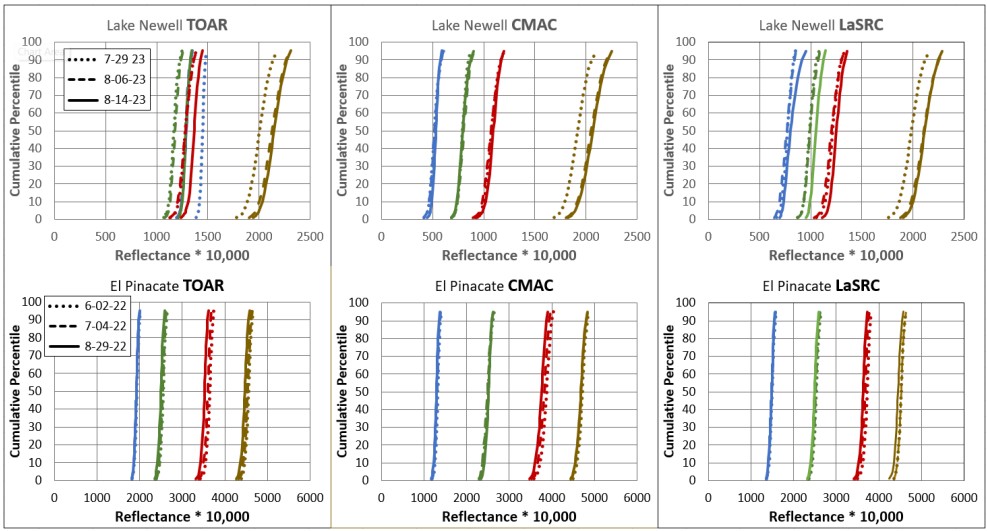

**Figure 4.** Extracted reflectance by treatment, with dot, dash, and solid patterns that identify each image evaluated by the two methods. Other than TOAR for Lake Newell, the bands in the graphs stack from left to right according to increasing wavelength: blue, green, red and NIR. Both AOIs have very low spectral dynamic range compared to Figure 2 and plot here as almost vertical lines.

The extremely low variability of the El Pinacate distributions in Figure 4 illustrates several trends. The CMAC-corrected bands have greater spacing and are positioned left of LaSRC. All three treatments portray red reflectance distributions as having less coherence.

The Lake Newell CMAC distributions are tighter than LaSRC. The unexpected discrepancy for NIR observed in the Lake Newell data was due to rain prior to the 14 August 2023 image, which was investigated and confirmed as described in Appendix E. Lake Newell LaSRC CDFs for the highest atmospheric effect 6 August 2023, 1044 (versus 920s for the other two dates) are displaced rightward. This discrepancy is an indication that the increase of Atm-I from 920s to 1044 resulted in under-correction by LaSRC; an interpretation based on the fact that atmospheric correction reduces the brightening effect of backscatter by moving the CDFs to the left. Hence under-correction results in the 6 August 2023 image being displaced rightward in relation to the corrected images from 29 July 2023 and 14 August 2023. The CMAC corrections of the visible bands were unaffected by Atm-I and maintain consistency and close agreement, as would be expected for the reflectance of midsummer shortgrass prairie when vegetation growth is essentially static.

Figure 5 presents the bandwise Lake Newell CDFs plotted with the surface reflectance points reconstructed from the SoCal AOIs identified through the workflow described earlier. For all bands, the reconstructed SoCal surface reflectance points of CMAC lie on the CDFs for Lake Newell. Many of the LaSRC points reconstructed in the same workflow, also lie on or close to the Lake Newell CDFs, partially corroborating that CMAC provides accurate surface reflectance estimates, though disagreeing with the LaSRC CDFs. Thus, for the Lake Newell comparisons, the null hypothesis that CMAC provides output equivalent to the surface reflectance surrogate dataset of SoCal is accepted. Judged by the data plots in Figure 5, any error between the SoCal and the Lake Newell datasets was slight.

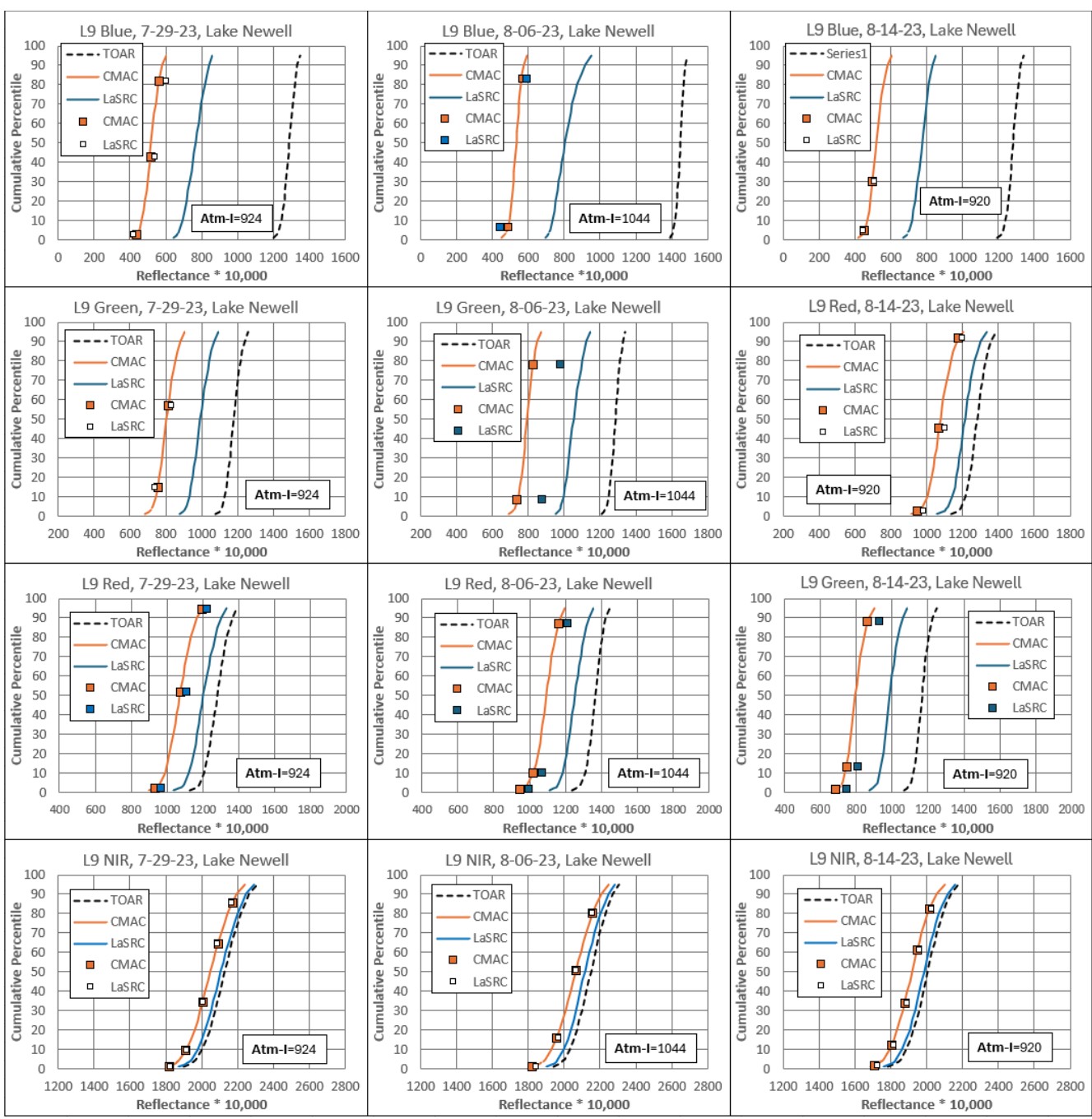

**Figure 5.** Lake Newell CDFs per band displayed with the points determined from the SoCal AOIs: Rochester (for 14 August 2023) and Fontana (for 29 July 2023 and 6 August 2023). The scaling of the x-axes is optimized to provide equivalent reflectance intervals for the same range per band. Where points overlie each other, the LaSRC SoCal points are depicted in white and are of smaller size.

In contrast to CMAC results, the points displayed for LaSRC reconstructed from the SoCal dataset TOAR values disagree with the Lake Newell reflectance distributions in all twelve graphic comparisons in Figure 5. The Lake Newell analysis was performed first. The analysis for the El Pinacate AOI was initiated to verify the same relationships for a different environment, one of profound aridity and almost no vegetation cover.

The El Pinacate data plotted in Figure 6 confirmed the results from the CMAC visible band in Figure 5 calculated from the shared SoCal TOAR reflectance values. The Fontana CMAC surface reflectance estimates lie close to the CMAC El Pinacate surface reflectance

distributions for blue, green, and red. The SoCal LaSRC points plotted closer to the CMAC distribution than to the LaSRC El Pinacate distributions. Figure 6 LaSRC NIR points plot differently than for Lake Newell (Figure 5), instead essentially lying on the reflectance distribution for El Pinacate TOAR, indicating that virtually no correction for NIR occurred.

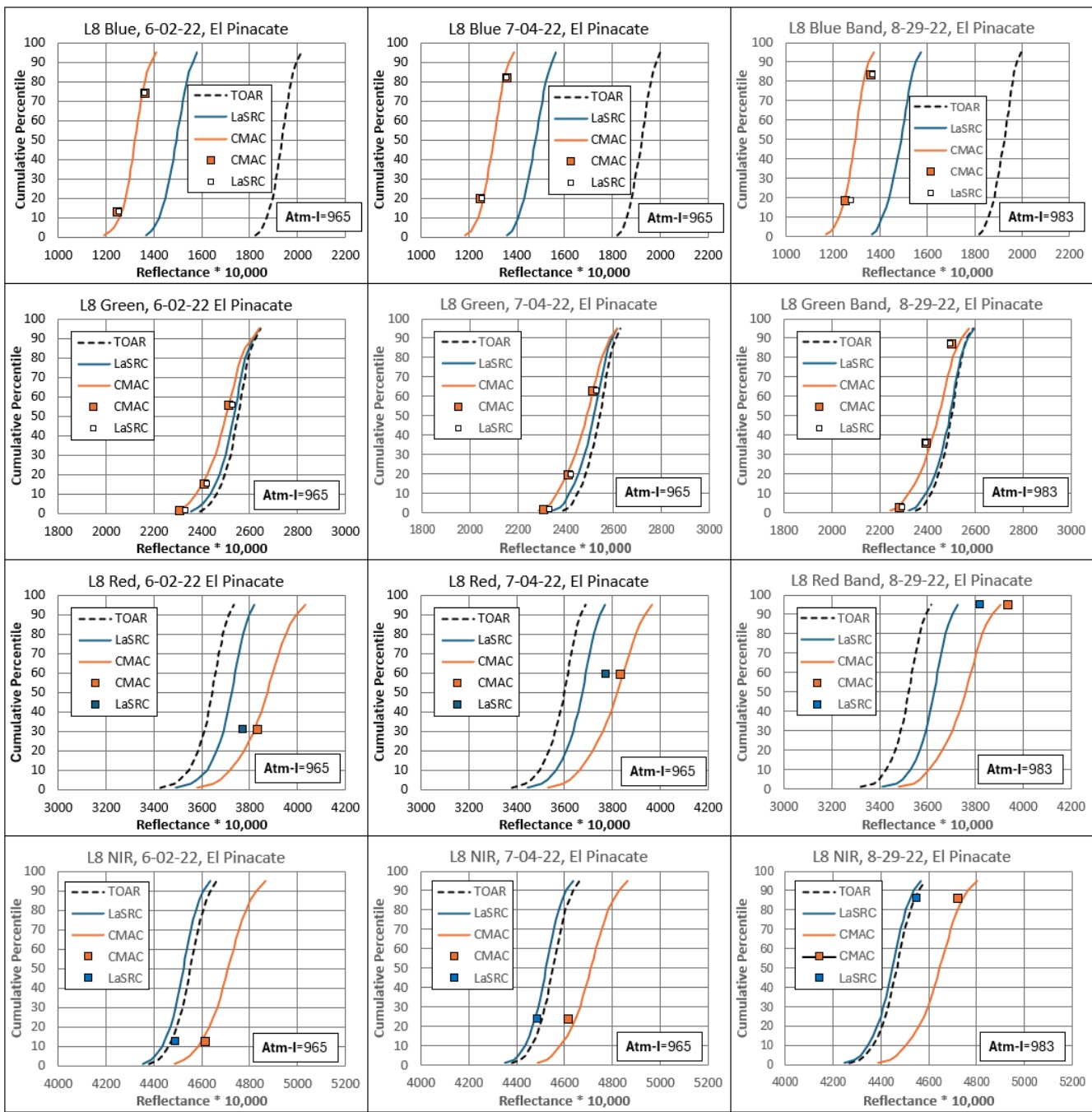

**Figure 6.** Per-band El Pinacate CDFs displayed with the points reconstructed from the SoCal AOI of Fontana. The scaling of the x-axes is optimized to provide equivalent reflectance intervals for the same range per band. Where points overlie each other, the LaSRC SoCal points are depicted in white and are of smaller size.

The error for surface reflectance estimation by CMAC and LaSRC was calculated by treating the surrogate SoCal reflectance values as true surface reflectance, presented in Tables 2 and 3. For CMAC, the Lake Newell and El Pinacate surface reflectance estimates

agree well with the SoCal surrogate true surface reflectance; CMAC error was low and almost evenly distributed between positive and negative values, and hence unbiased. CMAC results were comparable between the Lake Newell and El Pinacate datasets. The average absolute value of CMAC surface reflectance error did not exceed 1% for the blue band, which experienced the greatest error. The average absolute value of LaSRC error for the Lake Newell shortgrass prairie was severe, around 50% for the blue band. The error for LaSRC was lower for El Pinacate but still an order of magnitude greater than CMAC. The error for both CMAC and LaSRC decreased with increasing wavelength.

**Table 2.** Error calculated for surface reflectance estimation of three images of El Pinacate calculated from the average of absolute values. Each cell is the error in surface reflectance output from a TOAR input. This comparison included 21 individual comparisons across the four bands for each method.

| El Pinacate | CMAC 6-02-22 | 7-04-22 | 8-29-22 | Average Abs. Value | LaSRC 6-02-22 | 7-04-22 | 8-29-22 | Average Abs. Value |
|---|---|---|---|---|---|---|---|---|
| Blue | 1.3% −0.6% | 1.0% −0.9% | 1.0% −0.9% | 0.98% | 14.0% 12.3% | 13.9% 12.2% | 12.4% 12.1% | 13.23% |
| Green | −0.4% −0.3% 0.1% | 0.4% −0.3% 0.1% | −0.3% −1.0% 1.0% | 0.30% | 1.3% 1.3% 0.5% | 1.5% 1.1% 0.3% | 2.5% 3.3% 2.4% | 1.21% |
| Red | −0.3% | 0.3% | −0.8% | 0.80% | −2.1% | −2.2% | −2.5% | 1.86% |
| NIR | −0.5% | 0.1% | 0.6% | 0.40% | −1.0% | −0.1% | −0.6% | 0.59% |

**Table 3.** Error calculated for surface reflectance estimates for each band of three Lake Newell images. Overall averages were calculated from absolute values. Each cell is the error in surface reflectance output from a TOAR input. This comparison included 37 individual comparisons across the four bands for each method.

| Lake Newell | CMAC 7-29-23 | 8-06-23 | 8-14-23 | Average Abs. Value | LaSRC 7-29-23 | 8-06-23 | 8-14-23 | Average Abs. Value |
|---|---|---|---|---|---|---|---|---|
| Blue | 0.19% −1.29% −0.32% | −0.51% −0.68% — | 1.10% 0.21% — | 0.61% | 58.8% 40.11% 35.63% | 62.3% 48.73% — | 59.7% 46.82% — | 50.30% |
| Green | −0.32% −0.20% — | 0.26% −0.10% — | 1.32% −0.01% 0.37% | 0.37% | 26.91% 20.36% — | 12.57% 11.95% — | 18.67% 16.19% 13.30% | 17.14% |
| Red | −0.06% 0.00% −0.07% | 0.31% 0.33% −0.20% | 0.71% 0.07% 0.22% | 0.22% | 10.83% 8.50% 8.36% | 12.70% 10.35% 8.86% | 11.35% 9.54% 9.19% | 9.96% |
| NIR | −0.10% 0.01% −0.03% −0.02% −0.54% | 0.19% −0.01% 0.00% −0.37% | −0.04% −0.02% −0.03% 0.02% 0.00% | 0.09% | 3.10% 3.09% 2.74% 2.68% 2.84% | 3.10% 2.58% 2.31% 1.78% | 3.14% 3.11% 2.92% 2.68% 2.58% | 2.58% |

Sufficient data are presented in the Supplementary Materials to allow the interested reader to reconstruct and verify the workflow and the results. These include averages, interpolations, error calculations and spreadsheets. Values derived through this analysis are summarized in tables within Appendix D. Spreadsheets and shapefiles of the Fontana and Rochester AOIs are provided along with spreadsheets and shapefiles for the Lake Newell and El Pinacate AOIs. Cloud-based image browsing, selection, and CMAC correction and download of Landsat-8/9 and Sentinel-2 VNIR bands can be accessed through a link in Supplementary Materials.

## 4. Discussion

The CMAC surface reflectance estimates for Lake Newell and El Pinacate were within 99% agreement with the CMAC SoCal surrogate surface reflectance estimates in all 58 TOAR-based comparisons across the four VNIR bands (agreement was calculated as 100% minus the % error). The strong agreement for CMAC results between datasets from widely diverse environments validates the accuracy and reliability of CMAC processing and of its constituent assessment of atmospheric effect and the conceptual model-derived workflow that reverses it. Further corroborating CMAC accuracy is the observation that the LaSRC surface reflectance in Figures 5 and 6 for Lake Newell and El Pinacate lie closer to the CMAC distributions than to the LaSRC distributions, in many cases plotting atop the CMAC SoCal points.

The CMAC data demonstrate accuracy independent of the dynamic spectral range (highest minus lowest reflectance values): the SoCal AOIs had extremely wide ranges of values (Figure 2), while the spectral ranges for Lake Newell and El Pinacate were extremely narrow (Figure 4). For the clear to moderately hazy conditions examined here, the null hypothesis is accepted: CMAC analyses produced the same surface reflectance estimates from the same TOAR input under the same atmospheric conditions despite differences in the two terrestrial environments examined.

The LaSRC analysis demonstrated surface reflectance estimates with average agreement as low as 50%; hence, the null hypothesis is rejected—LaSRC was not reliable for estimation of surface reflectance across the two environments. This discrepancy may be related to the low dynamic spectral range of the Lake Newell and El Pinacate locations; however, this issue is more complicated because the same bandwise dynamic spectral ranges were comparable between these two experimental AOIs, but the degree of error for Lake Newell was about four times that of the El Pinacate error.

Atmospheric correction of satellite imagery by LaSRC, widely viewed as the state of the art in radiative transfer application for EO imagery, is proposed as the basis for smallsat atmospheric correction through a cross-calibration process with harmonized data from Landsat-8/9 and Sentinel-2 [8,9]. However, reliance upon LaSRC for smallsat applications can be expected to incorporate the same problems that reduce LaSRC accuracy. These problems include the loss of accuracy at higher levels of Atm-I that was found for LaSRC under conditions of increasing haze from wildfire [2], and from these results, the lack of reliable accuracy, hypothetically related to low-spectral-diversity environments.

CMAC is a unique pathway for atmospheric correction, and its testing here and in the previous two journal papers shows that its performance is more accurate over a wider range of atmospheric effects than Sen2Cor and LaSRC. Rather than delaying surface reflectance output while waiting for ancillary data, CMAC can process images immediately upon download from the satellite, because the only input for surface reflectance correction is from the image itself. Due to its robust and simple mathematical structure, CMAC can readily be calibrated for Smallsat application for any VNIR band combination due to its robust mathematical structure. CMAC will be adapted to correct data from hyperspectral sensors in a next-generation program that will include improving the accuracy of the Atm-I model, reliable overwater correction, and development/application of a calibration target and the technology to apply it under automation.

The greatest source of uncertainty in the CMAC workflow is the measure of atmospheric effect, Atm-I. While Atm-I can be shown to be far more sensitive than the ancillary data currently in use by LaSRC from MODIS [2], it was generated by a static assumption of reflectance of a reference crop rather than actual reflectance measurements. The key to this upgrade is extensive groundtruth. Likewise, extensive groundtruth will also permit spectral modeling to isolate and remove the specular reflectance component of water surfaces that could yield accurate water-leaving reflectance. Though CMAC generally performs well over water, the overarching effect of image geometry has not yet been characterized.

Calibration is the key for application of surface reflectance retrieval. This step is presently performed vicariously, requiring the use of master images of Sentinel-2 compared

to proxy images from smallsats. Even though these steps are automated, this program is inefficient because it requires visual assessment steps to ensure accuracy. Vicarious calibration can be replaced by a workflow that employs a well-engineered, constructed, managed, and monitored calibration target. Such a target is expected to yield greater precision, accuracy, and automation for CMAC calibration. The benefit of periodic automated calibration is that it allows detection and compensation of episodic in-orbit radiation-related sensor degradation [10].

## 5. Conclusions

This investigation confirms that CMAC provides reliable surface reflectance retrieval for all environments. When judged by the appearance of many varied images, CMAC accurately corrected all terrestrial environments, including deserts, arctic and alpine tundra, tropical and temperate forests, savannahs, grasslands, and farmland on six continents. CMAC also produced excellent results over the ocean, as can be seen in Appendix B Smallsats can be calibrated readily for direct application of CMAC without incorporating the additional uncertainty of ancillary data.

## 6. Patents

Currently, one CMAC patent is granted. Two additional patent applications are pending before the US Patent Trade Office, with one of these filed internationally through the Patent Cooperation Treaty (International Search Report—language approved as filed).

**Supplementary Materials:** The following supporting information can be downloaded from https://www.mdpi.com/article/10.3390/rs16122216/s1: "Live" Excel workbooks for the Fontana, Rochester, El Pinacate, and Lake Newell locations; collations for bandwise CDFs for the two experimental locations; and the shapefiles of the four locations to facilitate data extraction. Sentinel-2 and Landsat-8/9 images can be browsed, selected, and corrected by the user at https://strato.advancedremotesensing.com/app (accessed on 21 April 2024).

**Author Contributions:** D.G.; conceptualization, D.G. and T.R.; methodology, D.G. and T.R.; software, T.R. and D.G.; formal analysis, D.G.; data curation, T.R. and D.G.; visualization, D.G. and B.-C.G.; original draft, D.G.; review and editing, D.G., T.R. and B.-C.G.; resources, D.G. and B.-C.G.; project administration, D.G. All authors have read and agreed to the published version of the manuscript.

**Funding:** This research was funded by the U.S. National Science Foundation Small Business Innovation Research program through grants #1840196 and #1950746 and from personal funds of the lead author.

**Data Availability Statement:** Please see Supplementary Materials for accessing the analyzed data, shapefiles for the extraction of data, and the Cloud-based site for test correction and downloads of Landsat-8/9 images.

**Acknowledgments:** We thank the Landsat programs for the steady stream of free, high-quality imagery and the excellent documentation and celebrate Landsat's pathfinder role for imaging satellite technology. Our heartfelt thanks in memory of Thomas Loveland (deceased 13 May 2022), a central figure in Landsat applications, for his friendship, encouragement, and insights early in our R&D process.

**Conflicts of Interest:** Authors David Groeneveld and Tim Ruggles are employed by the company Advanced Remote Sensing Inc. The remaining author declares that the research was conducted in the absence of any commercial or financial relationships that could be construed as a potential conflict of interest.

## Appendix A. Brief Description of CMAC Workflow

*Appendix A.1. Atmospheric Effect Mapped as a Grayscale*

The first CMAC step employs the spatially discrete spectral band statistics from the scene as input to a model that assesses the atmospheric effect in images. The Atm-I model was developed using dense dark vegetation (DDV) measured by field spectrometry to

establish a standard value for a ubiquitous index crop. This crop was identified and sampled under a wide range of atmospheric and surface cover conditions that resulted in a robust scene-based statistical model [1]. Application of the Atm-I model generates a grayscale map that expresses the degree of effect across the image as a numerical scalar for the correction needed to return each pixel's TOAR to its original surface reflectance.

*Appendix A.2. Reversing the Mapped Atmospheric Effect to Deliver Surface Reflectance*

The second CMAC step reverses the atmospheric effect based on a conceptual model derived from an observed phenomenon that initially prompted CMAC development. For a hazy and clear pair of TOAR images over an AOI whose surface reflectance has remained relatively consistent, the CDFs will vary systematically, as shown in Figure A1. Imposition of increasing aerosol causes the distributions to rotate counterclockwise, and for decreasing aerosol, clockwise. This observation prompted development of a conceptual model as the basis for reversal of the atmospheric effect. To facilitate referencing this phenomenon, it was dubbed the "pinwheel effect". This conceptualization was valuable only as a first step in formulating the CMAC approach for atmospheric correction.

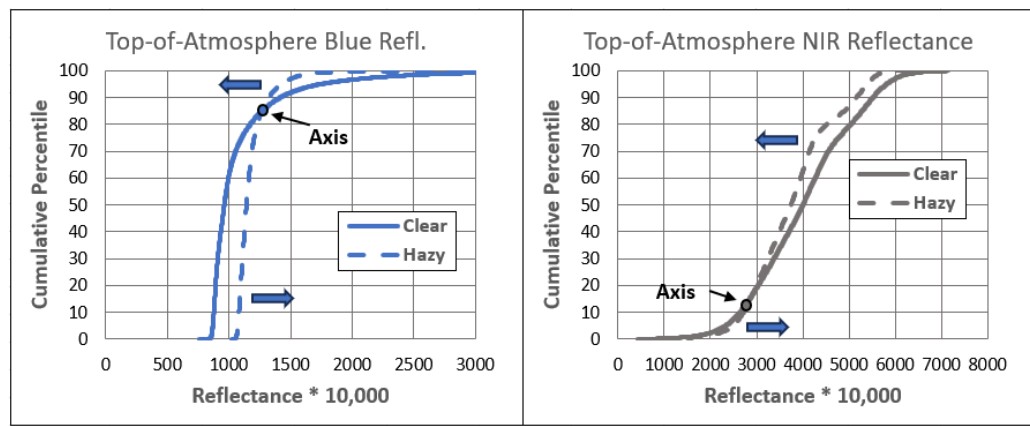

**Figure A1.** CDFs of two Sentinel-2 spectral bands extracted from an AOI with consistent surface reflectance across both image snapshots. Arrows show the direction of CDF rotation from increasing haze. This effect occurs in all VNIR bands.

In the pinwheel effect, increasing dark reflectance in response to increasing haze is due to aerosol backscatter of light. The decrease of bright reflectance is due to attenuation from absorption and diffuse shading by aerosol particles. When viewed as CDFs, changing levels of haze cause the distribution to rotate around an axis point where scatter and attenuation balance and the value remains unchanged. Subsequent observation indicated that this axis point migrates rightward with increasing Atm-I, hypothetically due to forward scatter as a property of the target's brightness. Target brightness influences the magnitude of the reflected energy and its interaction with atmospheric aerosol. In the context of EO, this forward scatter is the illumination of aerosol from below that increases in proportion to ground target brightness. Forward scatter is still somewhat hypothetical and is the subject of ongoing focus; however, it may hold the key to detecting and reversing the effect of specular reflectance over water. The Atm-I model is sensitive to forward scatter and performs well to remove specular reflectance from water (Appendix).

The application of data distributions was a key factor for the development of CMAC, since atmospheric correction seeks to return the range of TOAR pixel values to surface reflectance. CDFs are a robust means to approach the atmospheric correction problem, since individual pixel values could be correct in a distribution that is incorrect, but not vice versa. Additionally, the ranked position of any value within reflectance distributions remains the same through various treatments; hence, mathematically translated values can be identified afterward by their percentile position in the distribution. This feature enabled

finding equivalent properties for intercomparison of experimental data extracted from very different environments to test the null hypothesis.

The observed reflectance behavior of Figure A1 was expressed as a graphic model by inversion and adjustment of the well-known empirical line method [3]. The resulting linear model in Figure A2 has precedence in a 40-year-old paper by two prominent researchers (Figure A3).

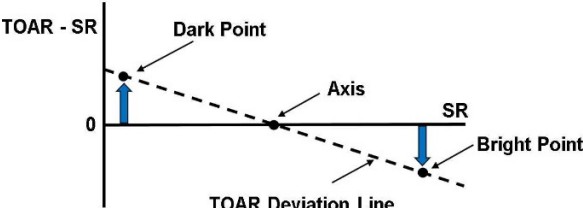

**Figure A2.** The CMAC conceptual model. Blue arrows indicate the rotational direction for increasing atmospheric effect. The dashed line represents all pixels, dark to bright, under one atmospheric effect for a single spectral band.

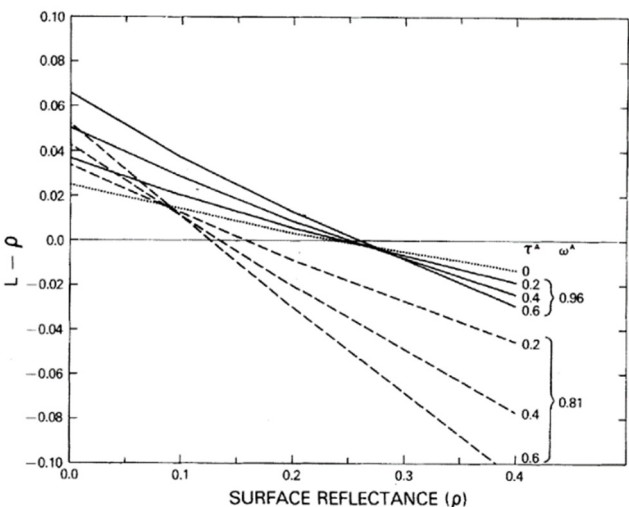

**Figure A3.** Figure 2, copied from Fraser and Kaufman, 1985 [11]. The solid lines represent common atmospheric aerosols and are equivalent to the dashed line in Figure 4. Dashed lines represent highly absorptive carbon particles to illustrate the importance of aerosol absorption upon reflectance.

The conceptual model of Figure 4 was translated into the CMAC equation that reverses atmospheric effect for each pixel of an image (Equation (A1)). The pinwheel effect of Atm-I in Figure 2 is represented by the upward/downward blue arrows in Figure 4, which results in linear deviation of surface reflectance due to the atmospheric effect. Slope and offset uniquely define any TOAR deviation line and are the parameters applied in Equation (A1) to reverse the atmospheric effect to retrieve surface reflectance. Slope and offset responses are unique for each sensor band and are determined through calibration. Presently, calibration is accomplished through image-to-image methods using a Sentinel-2 CMAC master calibration that was generated from several years of painstaking effort. Image-to-image calibration will eventually be replaced with a well-engineered and managed calibration target, providing increased accuracy and precision so that data from one to several sensor overpasses can be applied in an automated process. Automated recalibration will guard against episodic radiation-induced sensor changes, well known to occur in orbit [12].

$$SR = (TOAR - b)/(m + 1) \qquad (A1)$$

where m is slope and b is the offset of the TOAR deviation line.

## Appendix B. Annotated Spreadsheet Explaining Dataset Construction

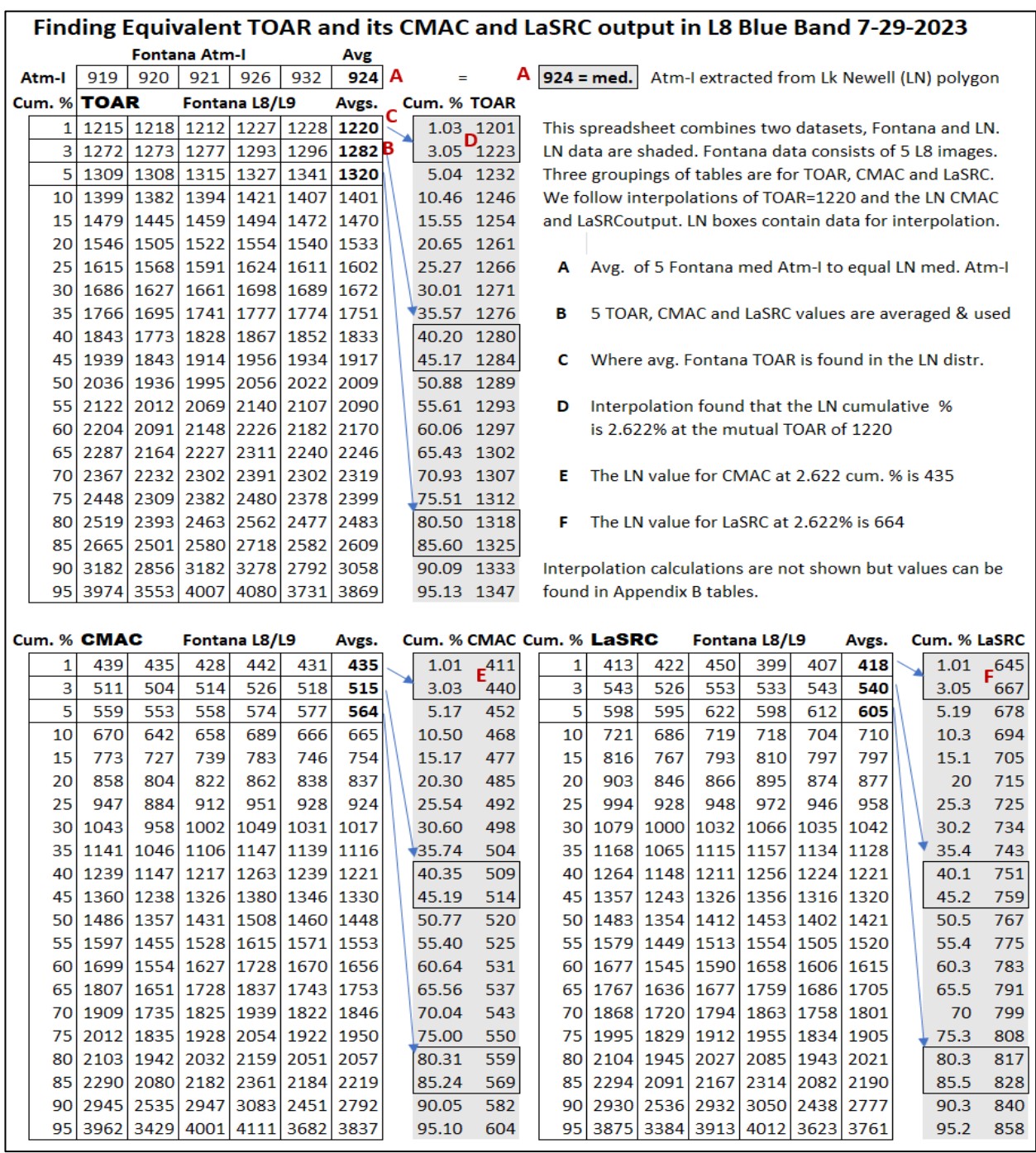

**Figure A4.** This annotated sample spreadsheet shows the calculation workflow for comparison of extracted L8 blue band data. Note that the TOAR, CMAC, and LaSRC tables are stacked vertically in the original spreadsheet but are rearranged here for ease of illustration. Tables of the five images are from the Fontana AOI, whose TOAR values correspond with the TOAR column for the Lake Newell AOI (shaded). Portions of the distribution are enclosed in boxes defining where the average TOAR of Fontana was found by interpolation of the Lake Newell distributions. The process began at **A** with the selection of five sequential images of Fontana, whose median Atm-I values were averaged and found to equal the median Atm-I of the Lake Newell AOI. The interpolations of the data in the shaded column boxes were performed graphically in the original spreadsheets, which are downloadable from the Supplementary Materials.

## Appendix C. El Pinacate Region Data Processed by CMAC and LaSRC for Visual Comparison: The 4 July 2022 Landsat-8 Image Displayed in TOAR, Atm-I, and CMAC and LaSRC-Corrected Views

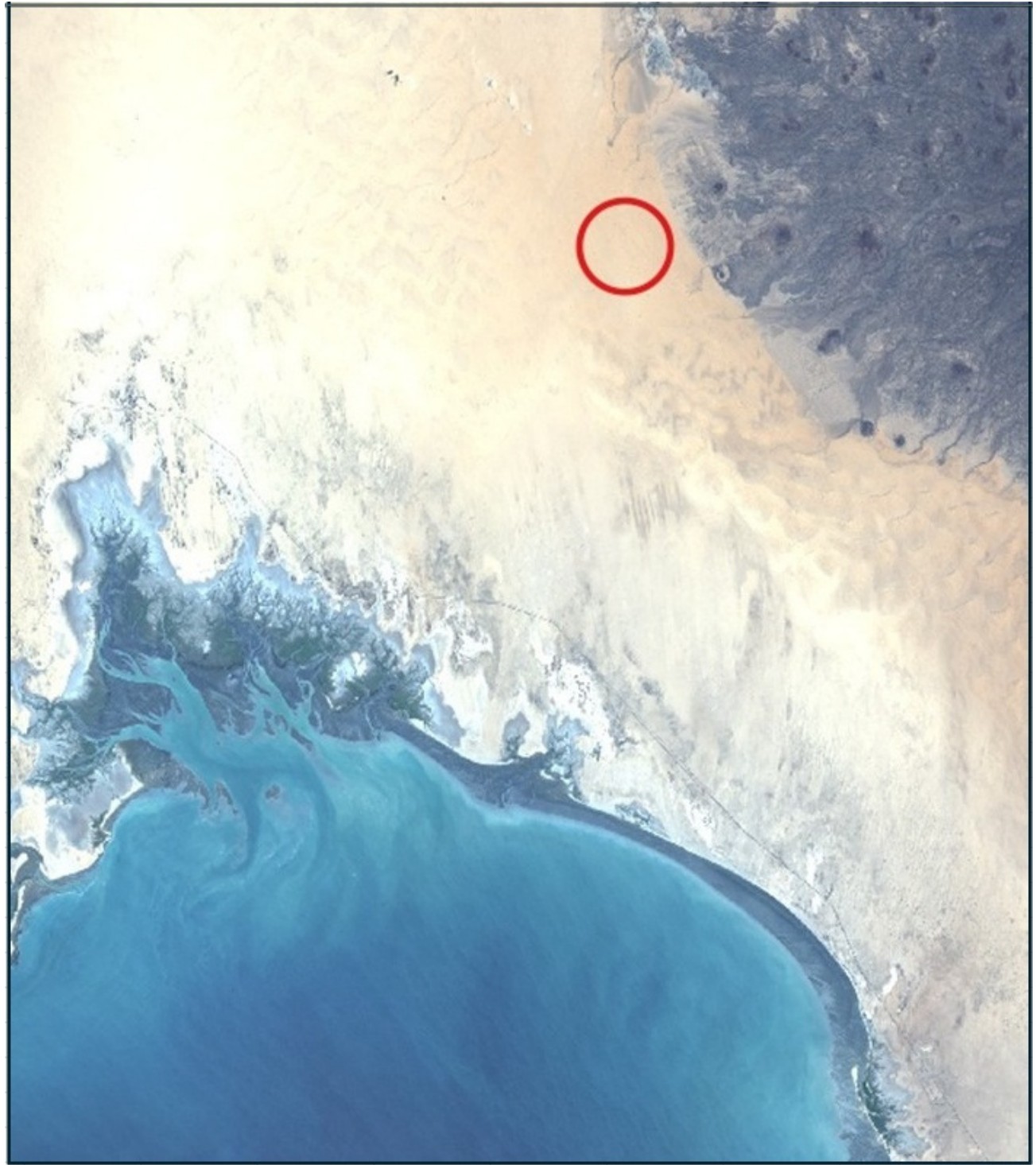

**Figure A5.** TOAR View of the El Pinacate region. The moderate level of haze obscures ground features over the desert. Light-colored features of the ocean result from a mix of entrained sediments in the water column and (hypothetically) from specular reflectance of the sky, This was influenced by wind, in evidence as streaks from the northwest. The AOI outline from which data were extracted is shown in red.

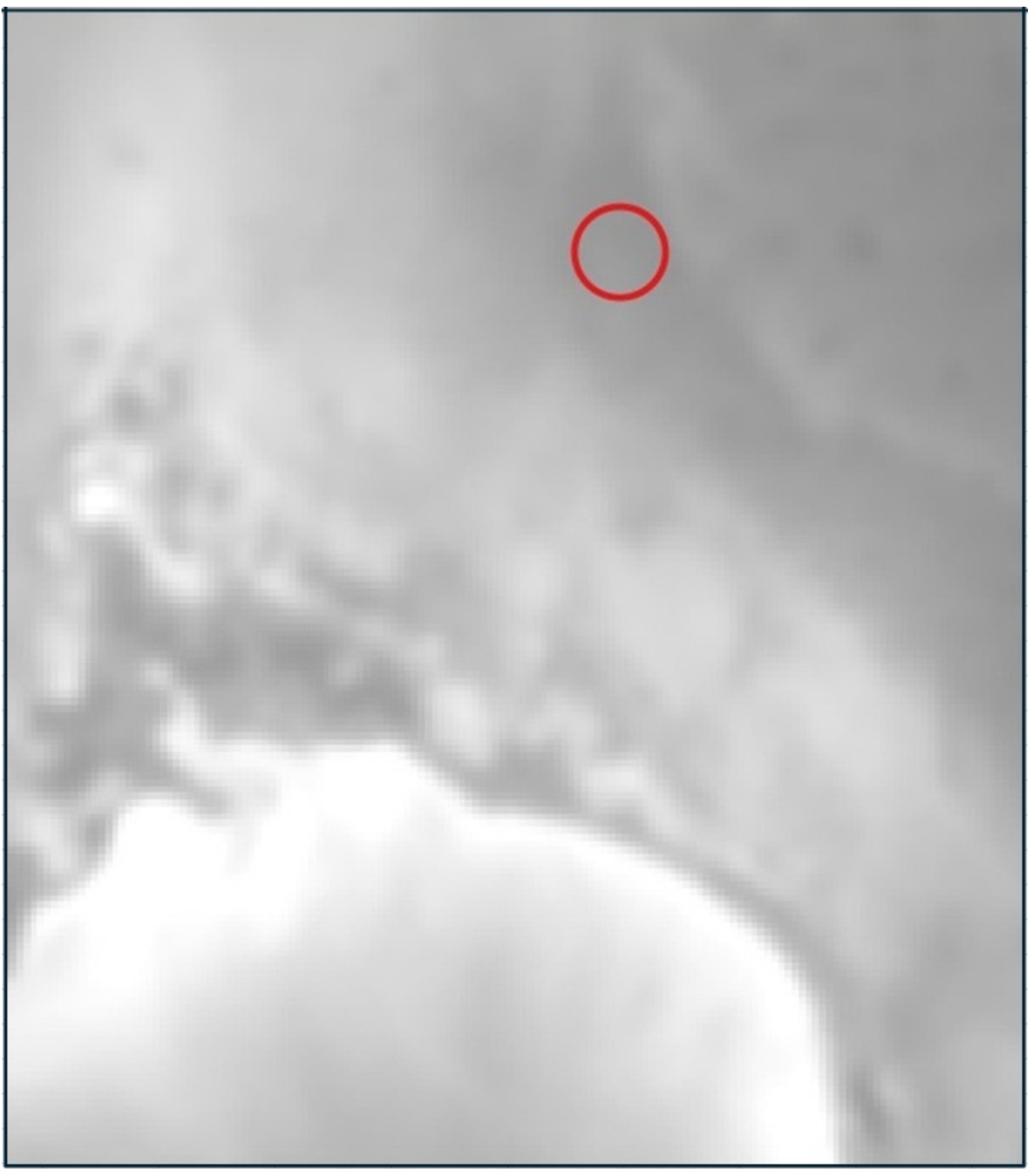

**Figure A6.** Atm-I grayscale view of the El Pinacate region. This atmospheric model output was applied to scale the degree of correction removing the atmospheric effect in the 4 July 2022 image: the brighter the response, the greater the correction. Of note is the brightness of the grayscale over the Sea of Corez, hypothetically induced by specular reflectance. Atm-I is a statistical representation of the atmospheric effect and, as such, has lower resolution than the original image; hence, the faint streaks from wind effects visible in the TOAR view are smeared in this view.

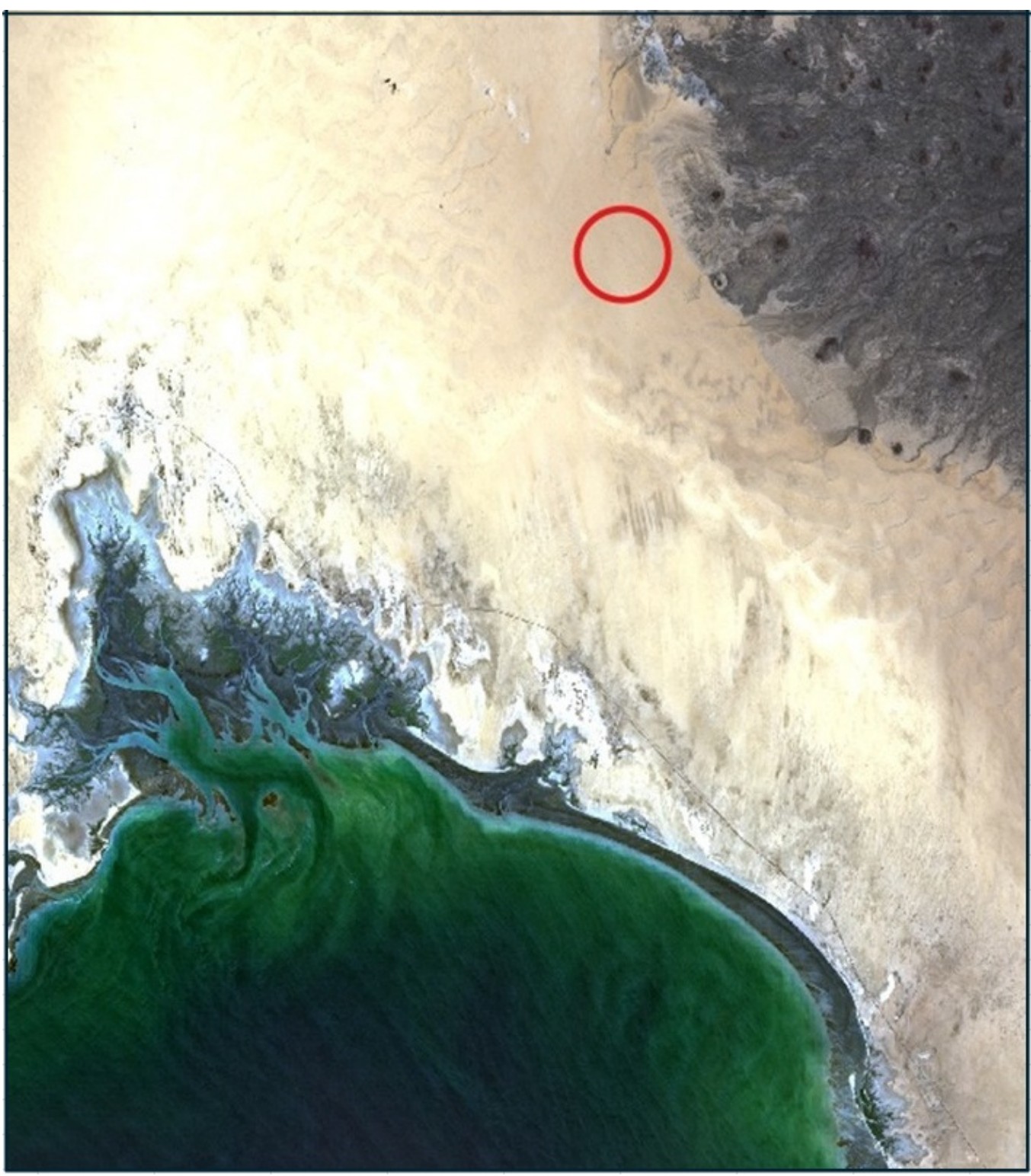

**Figure A7.** CMAC-corrected view with specular reflectance largely removed from the TOAR view of the ocean, patterns of entrained sediments and green-tinted water are now visible. Terrestrial features of windblown dunes and the complex hydrology surrounding the bay are visible after CMAC processing, though indistinct in the TOAR view. Future research is expected to define a relationship for the atmospheric statistical model (Atm-I) measurement of specular reflectance to enable reliable atmospheric correction over water.

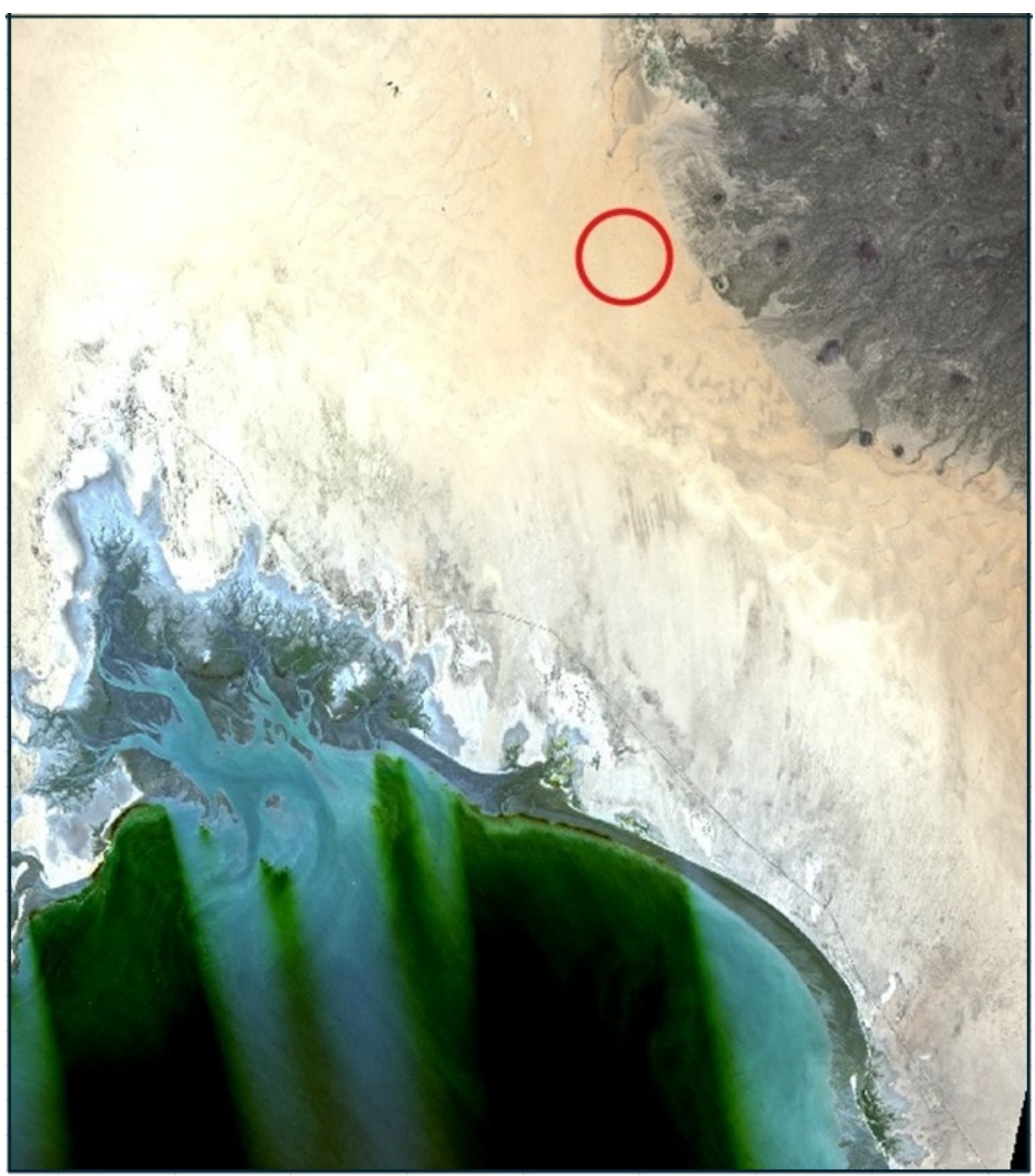

**Figure A8.** LaSRC-Corrected View. Like the CMAC correction, finer features of the image are visible after LaSRC correction. Image artifacts over the water are a common feature created by LaSRC correction. Such artifacts are also visible in the LaSRC view of Figure 1c in the main body of the text.

## Appendix D. Summaries of Spreadsheet Statistics

Data tables for three images for the El Pinacate and Lake Newell AOIs are provided below. Derivation of these data can be found in the spreadsheets available in the Supplementary Materials. Column 2 presents the TOAR shared values, common among the SoCal,

El Pinacate, or Lake Newell datasets. Corresponding percentiles for these shared TOAR values are presented in the third column. Interpolated reflectance values based on shared TOAR values are provided for SoCal CMAC (cols. 4 and 5) and LaSRC (cols. 7 and 8), and points are plotted Figures 5 and 6 and summarized in Tables 2 and 3 of the main text body from cols. 6 and 9.

**Table A1.** El Pinacate statistics.

| 1 | 2 | 3 | 4 | 5 | 6 | 7 | 8 | 9 |
|---|---|---|---|---|---|---|---|---|
| 6-02-22 L8 | Common TOAR | El Pinacate Percentile | Fontana CMAC | El Pinacate CMAC | CMAC % Error | QIA LaSRC | El Pinacate LaSRC | LaSRC % Error |
| Blue | 1880 | 13.0% | 1249 | 1266 | 1.3% | 1256 | 1431 | 14.0% |
| | 1964 | 74.1% | 1361 | 1352 | −0.6% | 1361 | 1528 | 12.3% |
| Green | 2399 | 1.5% | 2308 | 2301 | −0.3% | 2333 | 2362 | 1.3% |
| | 2482 | 15.0% | 2412 | 2403 | −0.4% | 2424 | 2456 | 1.3% |
| | 2559 | 55.8% | 2511 | 2513 | 0.1% | 2531 | 2542 | 0.5% |
| Red | 3613 | 31.0% | 3833 | 3822 | −0.3% | 3773 | 3693 | −2.1% |
| NIR | 4471 | 12.2% | 4616 | 4595 | −0.5% | 4489 | 4442 | −1.0% |
| 7-04-22 L8 | Common TOAR | El Pinacate Percentile | Fontana CMAC | El Pinacate CMAC | CMAC % Error | Fontana LaSRC | El Pinacate LaSRC | LaSRC % Error |
| Blue | 1880 | 20.0% | 1249 | 1262 | 1.0% | 1256 | 1430 | 13.9% |
| | 1964 | 82.0% | 1361 | 1348 | −0.9% | 1361 | 1527 | 12.2% |
| Green | 2399 | 1.8% | 2308 | 2300 | −0.4% | 2333 | 2367 | 1.5% |
| | 2482 | 19.6% | 2412 | 2405 | −0.3% | 2424 | 2451 | 1.1% |
| | 2559 | 62.9% | 2511 | 2515 | 0.1% | 2531 | 2538 | 0.3% |
| Red | 3613 | 59.4% | 3833 | 3844 | 0.3% | 3773 | 3689 | −2.2% |
| NIR | 4466 | 23.6% | 4616 | 4659 | 0.9% | 4489 | 4485 | −0.1% |
| 8-29-22 L9 | Common TOAR | El Pinacate Percentile | Fontana CMAC | El Pinacate CMAC | CMAC % Error | QIA LaSRC | El Pinacate LaSRC | LaSRC % Error |
| Blue | 1880 | 19.0% | 1249 | 1250 | 0.0% | 1256 | 1435 | 12.4% |
| | 1964 | 82.5% | 1361 | 1335 | −1.7% | 1361 | 1533 | 12.1% |
| Green | 2399 | 7.0% | 2308 | 2304 | −0.3% | 2333 | 2377 | 2.5% |
| | 2482 | 37.3% | 2412 | 2418 | 1.0% | 2424 | 2472 | 3.3% |
| | 2559 | 87.0% | 2511 | 2529 | 1.0% | 2531 | 2558 | 2.4% |
| Red | 3613 | 94.8% | 3833 | 3902 | −0.8% | 3773 | 3726 | −2.5% |
| NIR | 4541 | 86.0% | 4721 | 4743 | 0.5% | 4551 | 4522 | −0.6% |

**Table A2.** Lake Newell statistics.

| 7-29-23 L9 | Common TOAR | Lk Newell Percentile | Fontana CMAC | Lk Newell CMAC | CMAC % Error | Fontana LaSRC | Lk Newell LaSRC | LaSRC % Error |
|---|---|---|---|---|---|---|---|---|
| Blue | 1220 | 2.6% | 435 | 436 | 0.19% | 418 | 664 | 58.79% |
| | 1282 | 43.0% | 515 | 512 | −0.55% | 540 | 755 | 40.01% |
| | 1320 | 82.0% | 564 | 562 | −0.32% | 605 | 821 | 35.63% |
| Green | 1139 | 14.2% | 758 | 755 | −0.32% | 742 | 941 | 26.91% |
| | 1187 | 57.0% | 816 | 814 | −0.20% | 833 | 1002 | 20.36% |
| | 1087 | 2.3% | 932 | 932 | −0.06% | 964 | 1069 | 10.83% |
| | 1156 | 51.7% | 1075 | 1075 | 0.00% | 1111 | 1205 | 8.50% |
| Red | 1282 | 94.5% | 1197 | 1196 | −0.07% | 1225 | 1328 | 8.36% |

**Table A2.** *Cont.*

| | Common TOAR | Lk Newell Percentile | Fontana CMAC | Lk Newell CMAC | CMAC % Error | Fontana LaSRC | Lk Newell LaSRC | LaSRC % Error |
|---|---|---|---|---|---|---|---|---|
| NIR | 1812 | 1.4% | 1819 | 1817 | −0.10% | 1826 | 1883 | 3.10% |
| | 1913 | 9.8% | 1913 | 1913 | 0.01% | 1918 | 1977 | 3.09% |
| | 2001 | 34.4% | 2006 | 2005 | −0.03% | 2009 | 2064 | 2.74% |
| | 2088 | 64.3% | 2091 | 2091 | −0.02% | 2090 | 2146 | 2.68% |
| | 2243 | 85.5% | 2171 | 2170 | −0.04% | 2167 | 2229 | 2.84% |
| **8-06-23 L8** | Common TOAR | Lk Newell Percentile | Fontana CMAC | Lk Newell CMAC | CMAC % Error | Fontana LaSRC | Lk Newell LaSRC | LaSRC % Error |
| Blue | 1412 | 6.6% | 488 | 485 | −0.51% | 446 | 725 | 62.31% |
| | 1470 | 82.7% | 567 | 563 | −0.68% | 594 | 883 | 48.73% |
| Green | 1245 | 8.7% | 738 | 740 | 0.26% | 881 | 992 | 12.57% |
| | 1310 | 78.1% | 827 | 826 | −0.10% | 980 | 1097 | 11.95% |
| | 1249 | 1.5% | 943 | 946 | 0.31% | 997 | 1124 | 12.70% |
| | 1308 | 10.3% | 1018 | 1021 | 0.33% | 1072 | 1183 | 10.35% |
| Red | 1419 | 87.2% | 1164 | 1162 | −0.20% | 1216 | 1324 | 8.86% |
| NIR | 1940 | 1.0% | 1823 | 1827 | 0.19% | 1845 | 1903 | 3.10% |
| | 2056 | 15.9% | 1959 | 1958 | −0.01% | 1971 | 2022 | 2.58% |
| | 2150 | 50.8% | 2067 | 2067 | 0.00% | 2069 | 2117 | 2.31% |
| | 2230 | 80.1% | 2159 | 2151 | −0.37% | 2156 | 2201 | 2.10% |
| **8-14-23 L9** | Common TOAR | Lk Newell Percentile | Fontana CMAC | Lk Newell CMAC | CMAC % Error | Fontana LaSRC | Lk Newell LaSRC | LaSRC % Error |
| Blue | 1227 | 5.1% | 450 | 455 | 1.10% | 444 | 710 | 59.70% |
| | 1265 | 30.4% | 498 | 499 | 0.21% | 511 | 751 | 46.82% |
| Green | 1078 | 1.8% | 687 | 696 | 1.32% | 748 | 888 | 18.67% |
| | 1125 | 13.5% | 746 | 746 | −0.01% | 808 | 939 | 16.19% |
| | 1221 | 88.2% | 862 | 865 | 0.37% | 930 | 1053 | 13.30% |
| Red | 1171 | 2.6% | 948 | 955 | 0.71% | 981 | 1093 | 11.35% |
| | 1275 | 45.3% | 1069 | 1070 | 0.07% | 1100 | 1205 | 9.54% |
| | 1369 | 91.7% | 1177 | 1179 | 0.22% | 1201 | 1312 | 9.19% |
| NIR | 1808 | 1.8% | 1709 | 1708 | −0.04% | 1725 | 1780 | 3.14% |
| | 1897 | 12.2% | 1803 | 1803 | −0.02% | 1818 | 1874 | 3.11% |
| | 1967 | 34.0% | 1878 | 1877 | −0.03% | 1890 | 1946 | 2.92% |
| | 2032 | 61.1% | 1947 | 1947 | 0.02% | 1959 | 2012 | 2.68% |
| | 2096 | 82.2% | 2015 | 2015 | 0.00% | 2025 | 2077 | 2.58% |

## Appendix E. Why the Disconnect for the 14 August 2023 NIR Band CDF?

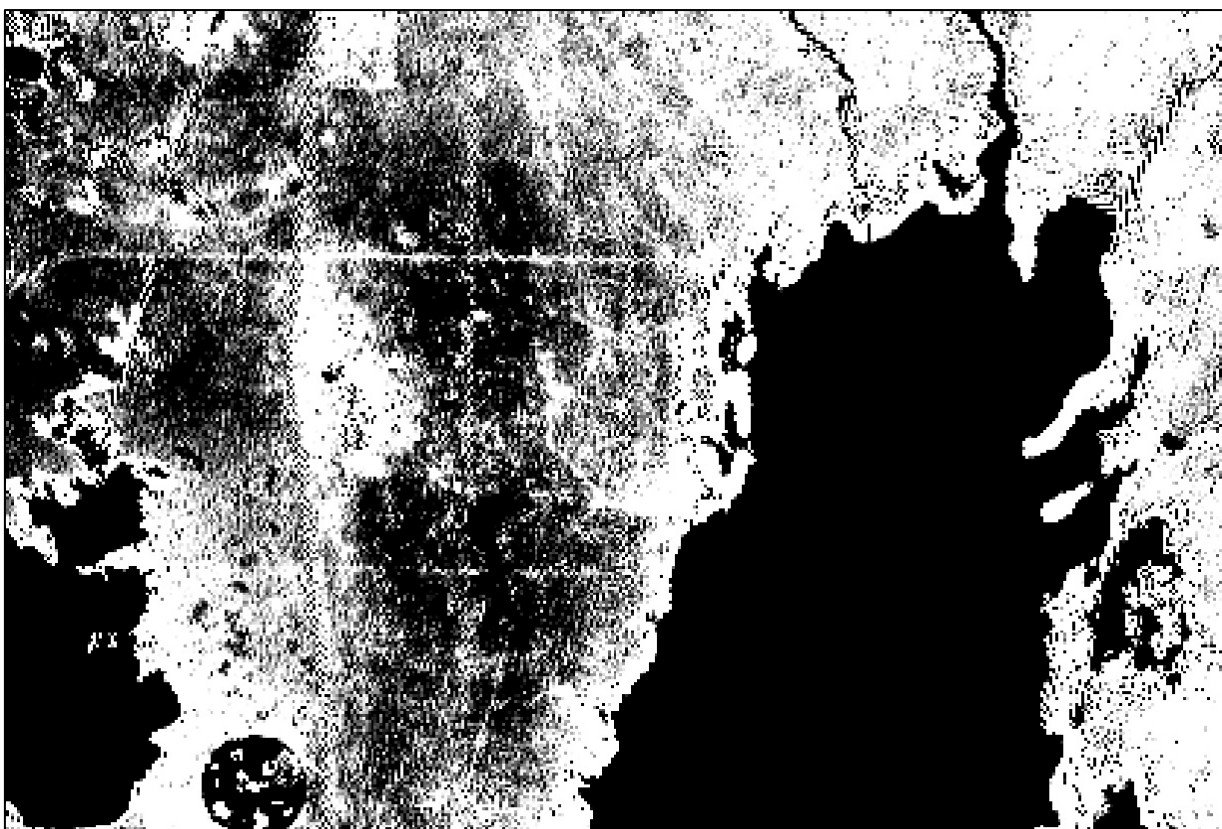

**Figure A9.** Change detection confirmed rainfall prior to 14 August, visible as a darkened smear to the west of Lake Newell. Figure 5 provides a color reference for the 6 August 2023 image. This analysis was performed to confirm the validity of the extracted NIR values to explain why the 14 August NIR results did not conform with the other two dates in Figure 4. The linear features that cross the area west of Lake Newell are gravel roads (confirmed on Google Earth) that drain rapidly and dry much quicker than the surrounding prairie. The prominence of these roads on the change detection image confirms that the darker area is not an atmospheric issue. Lakes appear black here because the TOAR reflectance was elevated due to haze (higher Atm-I); they are brighter on the 6 August image.

Steps in this image analysis were as follows:

1.  Subtract the 6 August 2023 NIR raster from the 14 August 2023 NIR raster. Minimum and maximum display values were selected to accentuate the display of wetter conditions guided by the magnitude of the differences visible for NIR in Figure 4.
2.  The resulting dark pattern was then compared against other variables across the AOI to ensure that this pattern was not explainable by some other image property—none were found.
3.  The analysis supports the conclusion that the AOI received rainfall from a localized thunderstorm, likely within several days prior to 14 August.

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
