# Peer review of "Landsat-8/9 Atmospheric Correction Reliability Using Scene Statistics"

_remotesensing, doi:10.3390/rs16122216_

Round 1

Reviewer 1 Report

Comments and Suggestions for Authors

This is an interesting paper introducing an alternative method for atmospheric correction.  CMAC is more of a "relative" scene correction algorithm using multiple scenes to come up with an atmospheric profile.  The simplicity certainly can be useful, especially when absolute estimates of SR are not needed and "clarity" of the image is most important.

Atmospheric correction continues to be challenging for all missions recording scientific quality satellite data.  CMAC could potentially be a problem solver if it proves to be effective.  To evaluate that, much more testing needs to be accomplished.  This study only evaluates it over 2 images.  Overall, the paper is good, but there are a few issues that need to be further explained before publishing.  The detailed points are in the attached PDF, but a summary of the major limitations are: 

1) The CMAC is not adequately explained in the paper.  It wasn't until the end, in the appendix, where it was explained.  This should be better explained with a paragraph or two in the introduction.

2) The SoCal results are heavily relied upon for error analysis, but there is no mention if those results where validated.  Additionally, they were created under "clean" conditions but are being applied to "widely diverse" conditions.  Furthermore, I fail to understand why ground truth measurements were not used for reference.  In general, I am skeptical of this dataset.

With more explanation of these two points the paper will be ready for publishing.   In the attachment, there are more edits and further explanations.

Author Response

Thank you for the thorough review of our paper.

Re: Number of images used. The entire count of images used must include the 31 Landsat images analyzed for the high spectral diversity SoCal location where the reflectance distribution was found to be stable over time. That analysis established that for high spectral diversity , dark to bright conditions, CMAC agreed extremely closely with LaSRC. Since CMAC is new it is compared in the paper to the accepted method, LaSRC, to benchmark CMAC performance. Specific to this analysis are six images that chosen for two very different locations that represented low spectral diversity environments.

The points brought up in your markup have been addressed within the manuscript.

  • Re: inadequate explanation of the method within the body of the paper rather than isolated in the Appendix (that was done to avoid cluttering the paper). Two paragraphs were added that provide a brief description of CMAC in the Methods selection. Appendix A remains for more detailed description and the reader is still referred to our other two journal papers that introduce the method in greater detail and describe an important secondary effect from forward scatter that is accommodated through calibration.
  • Note that the SoCal atmospheric conditions were within the same range as the two experimental sites, Lake Newell and El Pinacate. This allowed finding the exact estimate of atmospheric index between SoCal and the experimental sites. Ground truth measurements would be great, but two problems always will exist, scale and time. It is virtually impossible to get data at the scale necessary to relate to satellite imagery. The null hypothesis of “the same top-of-atmosphere reflectance under the same atmospheric condition will provide the same estimate of surface reflectance” is a thought-problem workaround for the lack of true groundtruth. However, the fact that the two datasets agree closely, their given TOAR input should then provide the same SR output that equals the original results from SoCal. Because LaSRC is the result of best efforts by NASA/Landsat, at least for relatively clean images, its output can be regarded as the best available estimate of surface reflectance. With regard to groundtruth, a few points of surface reflectance are of little value because it is the distribution that is important –points can be correct while the distribution is wrong. We are preparing an airborne system for data collection that will solve the problems for collecting surface reflectance measurements at scale.

Reviewer 2 Report

Comments and Suggestions for Authors

This manuscript focus on the atmospheric correction of Landsat-8/9 using scene statistics.

Generally,the method of this manuscript lacks innovation.

This manuscipt is more like an engineering report than an academic paper.

Author Response

Thank you for your review. I have made substantive changes to the document for your review. If there is something that still warrants criticism, please provide specific points I can address.

Reviewer 3 Report

Comments and Suggestions for Authors

This manuscript presents a novel calibration method for smallsat Earth observation platforms. The focus lies on a comparison between the authors' proposed CMAC method and the existing LaSRC software. The study finds the CMAC method to exhibit high stability and minimal errors, which is encouraging considering its recent development. The overall innovation and unexpected results are commendable.

However, the following aspects require further attention:

Literature Review: The introduction lacks a comprehensive discussion of prior work. Relevant studies include "Atmospheric correction of metre-scale optical satellite data for inland and coastal water applications" (Quinten Vanhellemont et al., 2018) and "Atmospheric correction of optical imagery from MODIS and Reanalysis atmospheric products" (Juan C. Jiménez-Muñoz et al., 2010). A more thorough review would strengthen the context and position the current research within the existing body of knowledge.

Key Assumptions: The paper's core argument heavily relies on assumptions made around line 80. The assumption that identical surface reflectance and atmospheric conditions will yield the same signal for the small satellite needs further clarification. In reality, varying satellite operation times and pointing directions inevitably lead to different signal corrections.  The authors should address how these factors are accounted for within the CMAC method.

Effectiveness of Typical Scenes: The analysis employs several key scenes obtained through remote sensing imagery. However, the paper would benefit from a more detailed discussion regarding the selection criteria and the overall effectiveness of these chosen scenes.

Applicability to Marine Environments: While the paper discusses water body reflection, the presented scenes primarily focus on land-based observations.  Further investigation into the applicability and stability of CMAC for predominantly marine reote sensing applications would be valuable.

LaSRC Blue Light Error: The significant error observed in the blue light portion of the results obtained using the LaSRC method warrants further explanation.  Investigating potential causes for this discrepancy would be beneficial.

By addressing these points, the authors can significantly strengthen their manuscript and contribute meaningfully to the field of small satellite Earth observation calibration.

Comments on the Quality of English Language

 Minor editing of English language required

Author Response

Thank you for your comments - Good insights.

Re: Literature review relative to prior work. I respectfully disagree relative to the literature review simply because CMAC is so different from all other approaches that other papers provide virtually no touch points for comparison. I downloaded the two papers you cited and confirmed that there is virtually nothing I can see that is relevant. That said, we are obviously standing on the shoulders of the Landsat program whose LaSRC software we have found to work very well if three conditions are met: (1) the degree of haze, i.e., atmospheric effect is not extreme (as we discovered in the paper cited as [2|), (2) that the dark-to bright spectral diversity is high, and (3) the target is land, not water. We have not downloaded the Jimenez-Munoz paper simply because we have noted that the CMAC method improves with increasing resolution, and we have not done work with decreasing resolution. Our interest has been restricted to serving smallsat atmospheric correction, though the method will work well for any optical EO satellite.  

Re: Key assumptions. The null hypothesis was developed as a test of reasonableness and as a simple and easily understood metric of image correction utility. Starting from surface reflectance as the standard for  atmospheric correction simplifies the problem and I cannot think of how to further clarify the problem since it is a logical measure that the same input under the same conditions provides the same output. CMAC does so in this paper and is now doing so for sets of many smallsat images as it has for hundreds of Sentinel-2 and Landsat-8/9 images, irrespective of path/row (i.e., looking at it toward the east or west), or whether as here for different L8 and L9 platforms (3 of each are included in the six images of the experimental Lake Newell and El Pinacate AOIs). The challenge is to estimate an accurate surface reflectance which CMAC does and so far, the results continue to support that the conceptual model in Appendix A mimics natural processes; also supported by the 1985 Fraser and Kaufman paper. I have some ideas why RadTran estimates from LaSRC fall short when any of the three conditions in the paragraph above are not met (but I have no data to back them up). A challenge we are preparing for is far off-nadir and extremely low solar horizon angle viewing, so we will take that up then.

Re: Effectiveness of Typical Scenes. The six Landsat scenes were part of our effort to test correct imagery from around the world. No criteria were imposed for the selection, otherwise. Lake Newell was found first and was investigated further after we corrected the notably hazy L8 image to clarity and compared to it LaSRC results that contained image artifacts prompting further investigation with contemporary L9 images to further explore image correction. El Pinacate was added on simply because it has even more profoundly low spectral diversity and has nearby  volcanic uplands for comparison.  I have added language that reflects this.

Re: Applicability to Marine Environments. Yes, this is a fascinating subject that I think can be solved through consideration of image geometry and differentiation of specular reflectance from atmospheric effects. So far, the atmospheric model (Atm-I) is doing a great job of getting rid of both (but not always). Please connect if you’d like to discuss further.

Re: LaSRC Blue error: This error is a problem in LaSRC, so I can hypothesize but can’t address how to fix it. It is not so much an issue with CMAC since the estimated blue band error from the present study was less than a percent. Our next-gen R&D should tighten up the Blue band even more.